# Analysis of Hospitalization Costs in Patients Suffering from Cerebral Infarction along with Varied Comorbidities

**DOI:** 10.3390/ijerph192215053

**Published:** 2022-11-16

**Authors:** Yongmei He, Sixuan Chen, Yongcong Chen

**Affiliations:** Institute of Social Medicine and Health Management, School of Public Health in Lanzhou University, Lanzhou University, Lanzhou 730030, China

**Keywords:** comorbidity, cerebral infarction, hospitalization cost, chronic disease

## Abstract

Objective: This study aimed to study the influence of comorbidities on hospitalization costs for inpatients with cerebral infarction. Methods: The data from the medical records pertaining to 76,563 inpatients diagnosed with cerebral infarction were collected from public hospital records for the period between 1 January 2020 and 30 December 2020 in Gansu Province. EpiData 3.1 software was used for data collation, and SPSS 25.0 was used for data analysis. Numbers and percentages were calculated for categorical variables, the chi-squared test was used to compare differences between groups, and multiple independent-sample tests (Kruskal–Wallis H test, test level α = 0.05) and multiple linear regression were used to analyze the influence of different types of comorbidity on hospitalization costs. Results: Among the 76,563 cerebral infarction inpatients, 41,400 were male (54.07%); the average age of the inpatients was 67.68 ± 10.75 years (the 60~80-year-old group accounted for 65.69%). Regarding the incidence of varied chronic disease comorbidities concomitant with cerebral infarction, hypertension was reported as the most frequent, followed by heart disease and chronic pulmonary disease. The average hospitalization cost of cerebral infarction inpatients is US $1219.66; the hospitalization cost increases according to the number of comorbidities with which a patient suffers (H = 404.506, *p* < 0.001); Regarding the types of comorbidities, the hospitalization cost of cancer was the highest, at US $1934.02, followed by chronic pulmonary disease (US $1533.02). Regarding the cost of hospitalization for combinations of comorbidities, cerebral infarction + chronic pulmonary disease was the most costly (US $1718.90), followed by cerebral infarction + hypertension + chronic pulmonary disease (US $1530.60). In the results of multiple linear regression analysis, cerebral infarction with chronic pulmonary disease had significant effects on hospitalization costs (β = 0.181, *p* < 0.001), drug costs (β = 0.144, *p* < 0.001) and diagnosis costs (β = 0.171, *p* < 0.001). Conclusions: Comorbidities are significantly associated with high hospitalization costs for cerebral infarction patients. Furthermore, relevant health departments should build preventative and control systems to reduce the risk of comorbidities, as well as to improve hospital clinical pathway management and to strengthen and refine the cost-control management of cerebral infarction from the perspective of comorbidities.

## 1. Introduction

Cerebral infarction (CI), also known as ischemic stroke, is an acute cerebrovascular disease characterized by sudden blockage of an artery, preventing blood supply to a specific area of the brain, constituting the main type of stroke [1,2]. In 2008, the World Health Organization defined comorbidity as the coexistence of two or more chronic conditions in the same individual [3,4]. With changes in lifestyle and the aggravation of ageing, stroke patients experience an increased risk of suffering from multiple chronic diseases. Suffering from comorbidities is a hallmark of stroke and places a heavy economic burden on society [5]. In North America, the cost of stroke ranges from US $3.6 billion in Canada to USD 34 billion in the US. In the Netherlands, the average hospital cost per patient for cerebral infarction is EUR 5328 (US $6845) [6]. The annual total expenditure for cerebral infarction was more than US $2898.6 million in China in 2011 [7]. The high medical costs as a result of comorbidities represent an important public health concern. The accumulating costs of different diseases vary greatly due to the presence of comorbidities [8]. Recently, many studies have focused on the medical costs of cerebral infarction patients regardless of their comorbidity status. However, there is a lack of research on the hospitalization costs from the perspective of comorbidities—especially in underdeveloped areas in the northwestern regions of China, such as Gansu Province. In this study, we focused on 13 chronic diseases, namely, hypertension, heart disease, diabetes, cancer, emotional and psychiatric disorders, asthma, chronic pulmonary disease, stomach or digestive diseases, arthritis or rheumatic disease, liver disease, kidney disease, memory-related diseases, and dyslipidemia. By accurately identifying the comorbidity patterns of cerebral infarction inpatients, and by exploring their influence on hospitalization costs, we can help relevant departments to formulate better medical insurance policies and health service guidelines, which will ultimately reduce the medical costs of patients with cerebral infarction and help allocate dedicated health resources reasonably. Hence, in this study, we thoroughly analyzed the associations between comorbidity status and hospitalization costs of cerebral infarction inpatients, providing us with the theoretical basis to better control the hospitalization costs of inpatients suffering with cerebral infarction.

## 2. Materials and Methods

### 2.1. Data Sources

The data were acquired from the “Direct Reporting System of Health Statistics Information Network” of the Health Commission of Gansu Province. We extracted data from the medical records pertaining to 76,563 inpatients with cerebral infarction who were discharged from public hospitals between 1 January 2020 and 30 December 2020 in Gansu Province. The Chinese National Bureau of Statistics showed that the average annual exchange rate between USD and RMB was 6.8996 RMB/USD in 2020 [9]. This study did not involve any private information on the patients or any ethical concerns.

Inclusion criteria:(1)According to the International Classification of Diseases, patients with the primary diagnostic disease code I63.900 were included.

Exclusion criteria:(1)Total hospitalization cost was zero;(2)Data were missing or incorrect (which could not be corrected);(3)Length of hospital stay was less than 1 day or more than 60 days.

### 2.2. Variables Extracted

(1)Basic inpatient information: gender, age, ethnicity, marital status, hospital level, and the patient’s chosen medical payment method;(2)Clinical information: discharge diagnosis, history of comorbidities, length of hospital stay, discharge and admission dates, and operation status;(3)Information relating to medical costs: total hospitalization costs, drug costs, diagnosis costs, treatment costs, costs of medical consumables, medical service costs, rehabilitation costs, and other costs.

### 2.3. Statistical Analysis

In this study, EpiData 3.1 software was used for data collation, whilst SPSS 25.0 was used for data analysis. The hospitalization costs, drug costs, and diagnostic costs were logarithmic and distributed in an approximately normal manner. Numbers and percentages were calculated for categorical variables. The chi-squared test was used to compare differences between groups. Multiple independent-sample tests (Kruskal–Wallis H test, test level α = 0.05) and multiple linear regression were used to analyze the influence of different types of comorbidity on hospitalization costs. In our study, all tests were two-tailed, and *p* < 0.05 indicated statistical significance.

## 3. Results

### 3.1. Basic Information of Inpatients

A total of 76,563 inpatients suffering from cerebral infarction were enrolled, including 41,400 males (54.07%) and 35,163 females (45.93%). The average age of the inpatients was 67.68 ± 10.75 years; 65.69% were 60~80 years old. Han ethnicity accounted for 94.01%, and married patients accounted for 82.22%. Only 12,519 patients (16.35%) did not have any comorbidities, while 64,044 patients (83.64%) suffered with comorbidities. The risk of comorbidity for patients aged 60–80 years was the highest, followed by those aged 40–60 years (χ^2^ = 0.080, *p* < 0.001); males had a significantly higher risk of stroke than females (χ^2^ = 0.023, *p* < 0.001). All differences were statistically significant (Table 1).

### 3.2. Association between Hospitalization Costs and the Number of Comorbidities

The average hospitalization cost of a patient suffering from a cerebral infarction was US $1219.66, and the average drug cost and diagnosis cost were US $391.77 and US $340.04, respectively. The average hospitalization cost of cerebral infarction patients without any comorbidities was estimated to be US $1174.31. On the other hand, the costs for patients suffering from one, two, three, and four or more chronic diseases were US $1224.41, US $1237.90, US $1247.30, and US $1322.27, respectively. Drug costs and diagnosis costs were the main components of hospitalization costs. The hospitalization costs (H = 404.506, *p* < 0.001), drug costs (H = 153.385, *p* < 0.001), and diagnosis costs (H = 491.911, *p* < 0.001) increased with the number of comorbidities. Statistically, these differences were significant, as shown in Table 2.

### 3.3. The Status of Hospitalization Costs in Different Types of Comorbidity

Among the types of comorbidity, the number of patients with comorbid hypertension (*n* = 45,626) was the highest, followed by heart disease (*n* = 27,646) and chronic pulmonary disease (*n* = 12,055); the hospitalization cost of cerebral infarction comorbid with cancer was the highest (US $1934.02), followed by chronic pulmonary disease (US $1533.02), asthma (US $1488.22), memory-related diseases (US $1439.01), and diabetes (US $1391.31) (Table 3).

### 3.4. The Status of Hospitalization Costs in Different Combinations of Comorbidities

The 20 most frequently related comorbidities accompanying cerebral infarction are listed in Figure 1. Based on the average hospitalization costs, the top seven were reported to be (1) cerebral infarction + chronic pulmonary disease (US $1718.90), (2) cerebral infarction + hypertension + chronic pulmonary disease (US $1530.60), (3) cerebral infarction + hypertension + diabetes (US $1375.56), (4) cerebral infarction + heart disease + chronic pulmonary disease (US $1318.41), (5) cerebral infarction + hypertension + diabetes + heart disease (US $1304.49), (6) cerebral infarction + diabetes (US $1292.39), and (7) cerebral infarction + hypertension + heart disease + kidney disease (US $1283.33) (Figure 1).

### 3.5. Multiple Linear Regression Analysis of the Impacts of Comorbidities on Hospitalization Costs for Cerebral Infarction Patients

Among the chronic disease types, patients suffering with a cerebral infarction along with chronic pulmonary disease had the greatest effect on hospitalization costs (β = 0.181, *p* < 0.001), followed by cancer (β = 0.156, *p* < 0.001). Cerebral infarction along with emotional and psychiatric disorders had the greatest effect on drug costs (β = 0.370, *p* < 0.001), followed by chronic pulmonary disease (β = 0.144, *p* < 0.001). Cerebral infarction along with chronic pulmonary disease had the greatest effect on diagnostic costs (β = 0.171, *p* < 0.001), followed by emotional and psychiatric disorders (β = −0.149, *p* < 0.001) (Table 4).

## 4. Discussion

Among the types of chronic disease, hypertension and heart disease were more prone to occurring in patients alongside cerebral infarction. Related research showed that hypertension was a hallmark comorbidity of stroke. Moreover, large-artery atherosclerosis (LAA) stroke and small-vessel occlusion (SVO) stroke were closely related to hypertension. Moreover, studies have shown that cardioembolic stroke is associated with atrial fibrillation, valvular heart disease, and ischemic heart disease [10,11]. Hypertension transmits pulsating and turbulent blood flow to the brain’s microcirculation, which increases endothelial dysfunction and atherosclerosis, ultimately promoting the development of stroke. Heart disease and cerebral infarction have common risk factors and a similar underlying pathogenesis; for example, atrial fibrillation (AF) increases the risk of patients suffering from cerebral infarction also suffering from heart disease [12]. Our study found that patients suffering from chronic pulmonary disease have an increased risk of developing cerebral infarction (*β* = 0.181, *p* < 0.001). Relevant studies have shown that cerebral infarction patients are often associated with stroke-related pneumonia in the clinic, and the rate of comorbidity has been reported as high as 10~47% [13]. Chronic obstructive pulmonary disease can increase the risk of atherosclerotic stroke, which is an independent risk factor of cerebral infarction [14]. Patients presenting with cerebral infarction often also manifest with symptoms of dysphagia and a weakened cough reflex. The reason for this could be because secretions from the trachea become difficult to cough out after a stroke, resulting in their deposition in the respiratory tract and consequent chronic pulmonary infections [15,16,17]. A Danish study found that almost half of patients with occult lung cancer were identified three months (or more) after suffering from a stroke (notably, smoking was a common risk factor for both) [18]; therefore, it is necessary to identify patients who are at a high risk of suffering from chronic diseases that are related to cerebral infarction—especially hypertension, heart disease, and chronic lung diseases. It is also critical to strengthen the prevention and control of cerebral infarction and associated comorbidities.

In terms of medical costs, the hospitalization costs of patients suffering from a cerebral infarction along with one or more comorbidities were significantly higher than those not suffering with a comorbidity. The greater the number of comorbidities, the higher the hospitalization costs of cerebral infarction patients. Studies have shown that the superposition, coexistence, and combination of multiple chronic diseases can exacerbate a patient’s disease, leading to additional overutilization of health services and the consumption of medical resources, ultimately increasing medical expenditures [19]. Relevant studies have shown that the numbers of coexisting chronic diseases are associated with the levels of health service utilization and health expenditure. The duration of hospital stay and health expenditure increased by 1.73- and 1.34-fold for each additional comorbid chronic disease, respectively [20]; this represents an extremely significant rise in costs. Multiple studies from France have shown that when multiple chronic diseases coexist, the interaction between diseases produces clear super-accumulation, resulting in significantly increased healthcare-related costs [21]. American studies have also shown that suffering from comorbidities is an important factor that affects medical costs. The comorbidity index increases hierarchically, i.e., the higher the comorbidity index, the higher the hospitalization costs [22]. Therefore, controlling the numbers of comorbidities in patients with cerebral infarctions is crucial to reducing the consumption of medical resources and the costs of hospitalization.

Among the types of comorbidities commonly associated with cerebral infarctions, the hospitalization cost of cancer was the highest, followed by chronic pulmonary disease.

Among combinations of comorbidities, cerebral infarction + chronic pulmonary disease resulted in the highest hospitalization costs; thus, chronic lung disease is a disease that cannot be ignored in the context of our discussion. In addition, although the number of cerebral infarction patients who also suffered with hypertension and heart disease was the greatest, the cost of their hospitalization was relatively low, which might be closely related to Chinese medical insurance policies that are implemented in relation to diabetic and hypertensive drugs. As part of China’s healthcare reformation, hypertensive drugs have been uniformly included in Medicare payments, and the associated reimbursement rate is now above 50%. The implementation of the “basic medical insurance + serious illness insurance system” policy has also greatly reduced the out-of-pocket payment ratio of hospitalization costs for cerebral infarction patients also suffering with hypertension and heart disease [23,24]. In contrast, the proportion of treatment drugs related to cancer and chronic pulmonary disease is low, and the proportion of out-of-pocket expenses of these patients is high, so the hospitalization costs of cerebral infarction combined with cancer or chronic pulmonary disease are higher. Studies have shown that the direct economic burden of chronic obstructive pulmonary disease patients accounts for almost 1/3 of the average annual household income of patients in China [25]. In Italy, the average annual cost per patient to treat chronic obstructive pulmonary disease is EUR 3911.70 [26]. Studies have confirmed that patients suffering with stroke and pulmonary disease have a high risk of comorbidities. The average hospital stay for a stroke patient with a comorbid chronic lung infection was 2.6 times longer than that of a patient without a chronic lung infection (13 days vs. 5 days); moreover, the average annual medical hospitalization cost was 3.5 times higher (US $21,043 vs. US $6206) [27,28,29]. Based on the aforementioned information, we believe that it is necessary to implement a reasonable increase in the set reimbursement of medical insurance that is related to medications for cerebral infarction and chronic pulmonary disease drugs. It is also necessary to introduce a variety of effective measures to reduce the hospitalization costs of patients suffering from comorbidities.

Multiple linear regression analyses between hospitalization costs and types of comorbidities related to cerebral infarction revealed that chronic pulmonary disease had a significant influence on hospitalization and drug costs. On the one hand, chronic pulmonary disease is a progressive and irreversible disease, manifesting as a persistent cough or as asthma, which greatly affects patients’ quality of life, resulting in higher clinical medical needs as well as drug dependence. On the other hand, chronic pulmonary disease is characterized by polypharmacy and pharmacotherapeutic complexity. A patient’s medication compliance, drug-related side effects, and drug interactions all result in increased healthcare costs [30]. Cerebral infarction comorbid with chronic pulmonary disease not only increases the length of hospitalization but also results in high consumption of medical resources, which leads to high hospitalization costs. At present, the policies related to cost-control and intervention strategies used by the Chinese health authorities with respect to cerebral infarctions and their related comorbidities are not perfect. Hence, the relevant health departments should focus more of their attention on the medical costs of cerebral infarctions occurring alongside chronic pulmonary diseases, while also attempting to reduce the risk of cerebral infarction combined with chronic lung diseases.

## 5. Conclusions

Comorbidities are significantly associated with high hospitalization costs for cerebral infarction patients. Furthermore, relevant health departments should build prevention and control systems to not only reduce the risk of patients developing comorbidities but also improve the clinical management pathways and hospitalization cost-control system, as well as better refining the cost-control management of cerebral infarction from the perspective of comorbidities.

## Figures and Tables

**Figure 1 ijerph-19-15053-f001:**
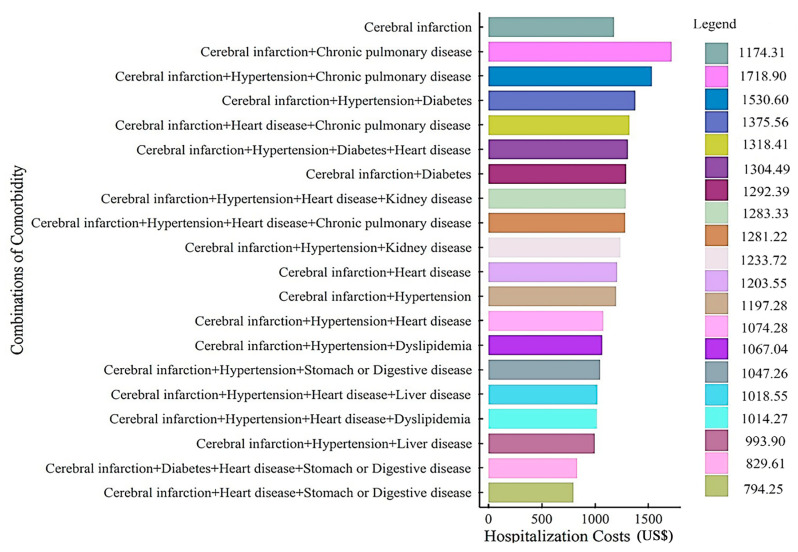
Status of hospitalization costs for different combinations of comorbidities with cerebral infarction.

**Table 1 ijerph-19-15053-t001:** Sociodemographic characteristics of discharged patients with cerebral infarction (*n* (%)).

Variable	No Comorbidity Group	Comorbidity Group	Total	χ^2^	*p*
Gender				0.023	<0.001
Male	7088 (9.26)	34,312 (44.82)	41,400 (54.08)		
Female	5431 (7.09)	29,732 (38.83)	35,163 (45.92)		
Age, years				0.080	<0.001
19	273 (0.36)	746 (0.97)	1019 (1.33)		
40	3684 (4.81)	13,896 (18.15)	17,580 (22.96)		
60	7636 (9.97)	42,655 (55.72)	50,291 (65.69)		
80	926 (1.21)	6747 (8.81)	7673 (10.02)		
Ethnicity				−0.036	<0.001
Han	11,539 (15.07)	60,504 (79.03)	72,043 (94.10)		
Ethnic minorities	980 (1.28)	3540 (4.62)	4520 (5.90)		
Marital status				0.030	<0.001
Unmarried	1945 (2.54)	8955 (11.70)	10,900 (14.24)		
Married	10,310 (13.47)	52,637 (68.75)	62,947 (82.22)		
Divorced/widowed	264 (0.34)	2452 (3.20)	2716 (3.54)		
Hospital level				0.198	<0.001
First-level hospital	897 (1.17)	1279 (1.67)	2176 (2.84)		
Secondary hospital	8019 (10.47)	28,663 (37.44)	36,682 (47.91)		
Tertiary hospital	3603 (4.71)	34,102 (44.54)	37,705 (49.25)		
Total	12,519 (16.35)	64,044 (83.65)	76,563 (100.00)		

**Table 2 ijerph-19-15053-t002:** The status of costs in different numbers of comorbidities for cerebral infarction patients (US $).

Number of Comorbidities	Hospitalization Costs	Drug Costs	Diagnosis Costs	Treatment Costs	Medical Consumables Costs	Medical Service Costs	Nursing Costs	Rehabilitation Costs	Other Costs
0 (*n* = 12,519)	1174.31	352.88	314.30	188.61	115.76	67.00	46.37	66.36	23.02
1 (*n* = 21,614)	1224.41	398.26	338.22	176.28	115.12	90.59	60.43	40.92	4.58
2 (*n* = 23,146)	1237.90	402.38	330.07	180.70	139.68	85.88	58.31	33.21	7.69
3 (*n* = 16,014)	1247.30	393.84	364.97	169.61	120.55	91.14	59.12	45.47	2.59
≥4 (*n* = 3270)	1322.27	414.56	395.28	188.15	114.46	89.65	66.36	50.83	2.98
Total	1219.66	391.77	340.04	178.66	123.27	85.48	57.51	37.19	5.74
*H*	404.506	153.385	491.911	55.347	26.427	1236.598	427.936	258.721	379.125
*p*	<0.001	<0.001	<0.001	<0.001	<0.001	<0.001	<0.001	<0.001	<0.001

**Table 3 ijerph-19-15053-t003:** The status of costs in different types of comorbidities for cerebral infarction patients (US $).

Type of Comorbidity	*n*	Drug Costs	Treatment Costs	Diagnosis Costs	Material Costs	Medical Service Costs	Nursing Costs	Rehabilitation Costs	Other Costs	Hospitalization Costs	Rank
Cancer	3253	576.05	316.59	468.05	306.84	113.42	88.74	60.05	4.28	1934.02	1
Chronic pulmonary disease	12,055	503.90	224.60	410.02	144.62	102.97	96.29	46.06	4.57	1533.02	2
Asthma	412	508.30	225.25	445.58	118.64	96.88	68.01	23.88	1.69	1488.22	3
Memory-related disease	1557	482.22	170.17	383.08	103.72	98.66	112.97	84.69	3.51	1439.01	4
Diabetes	11,642	439.92	196.73	409.36	127.19	84.87	65.40	62.03	5.82	1391.31	5
Kidney disease	6719	418.68	184.62	402.24	96.40	96.74	67.21	64.14	2.32	1332.36	6
Arthritis or rheumatism disease	1815	346.40	204.91	316.20	186.24	91.63	46.99	29.42	2.56	1224.34	7
Hypertension	45,626	394.25	172.19	339.44	116.05	87.12	56.28	41.12	5.82	1212.28	8
Heart disease	27,646	370.42	156.70	329.04	136.28	85.60	52.67	32.31	2.43	1165.44	9
Liver disease	7436	362.28	142.46	361.92	68.73	85.32	53.59	55.08	2.27	1131.64	10
Dyslipidemia	5565	342.61	152.28	336.93	64.16	75.24	35.24	34.32	2.50	1043.28	11
Stomach or digestive disease	6826	319.16	138.54	285.17	91.57	93.86	47.74	26.20	2.50	1004.73	12
Emotional and psychiatric disorders	2951	274.97	130.63	284.34	52.47	122.38	45.02	28.72	1.60	940.12	13

**Table 4 ijerph-19-15053-t004:** Multiple linear regression analysis of the impacts of comorbidities on hospitalization costs.

Variable	Hospitalization Costs				Drug Costs				Diagnosis Costs			
	Beta	t	95% CI		Beta	t	95% CI		Beta	t	95% CI	
(Constant)	7.356 **	504.918	7.385	7.328	5.685 **	244.702	5.731	5.640	6.860 **	365.512	6.896	6.823
Gender (Ref. = female)	−0.048 **	−12.684	−0.041	−0.056	−0.057 **	−9.398	−0.045	−0.069	−0.029 **	−5.891	−0.019	−0.039
Age (Ref. = 60)												
19	0.148 **	8.985	0.180	0.115	−0.014	−0.551	0.037	−0.065	0.073 *	3.462	0.115	0.032
40	0.022 **	4.790	0.031	0.013	0.007	0.987	0.022	−0.007	−0.001	−0.182	0.011	−0.013
80	0.050 **	7.822	0.063	0.038	0.073 **	7.131	0.093	0.053	0.053 **	6.290	0.069	0.036
LN length of hospital stay	0.535 **	129.207	0.543	0.527	0.822 **	122.883	0.835	0.809	0.163 **	30.466	0.174	0.153
Hospital level (Ref. = Tertiary hospital)												
First-level hospital	−0.042 **	−3.771	−0.020	−0.064	−0.042 *	−2.313	−0.006	−0.078	−0.109 **	−7.446	−0.081	−0.138
Secondary hospital	0.466 **	116.274	0.474	0.458	0.405 **	63.806	0.417	0.392	0.586 **	113.117	0.596	0.575
Payment (Ref. = medical insurance)												
Public expense	0.029 *	3.003	0.047	0.010	−0.124 **	−8.304	−0.095	−0.154	0.044 **	3.641	0.068	0.020
Out-of-pocket	0.225 **	16.228	0.252	0.197	0.125 **	5.708	0.167	0.082	0.189 **	10.602	0.224	0.154
Other	0.067 **	4.234	0.099	0.036	−0.156 **	−6.215	−0.107	−0.205	0.014	0.693	0.055	−0.026
Type of comorbidity (Ref. = cerebral infarction)												
Hypertension	−0.040 **	−10.292	−0.033	−0.048	−0.025 **	−3.954	−0.012	−0.037	−0.077 **	−15.205	−0.067	−0.087
Diabetes	0.069 **	12.976	0.080	0.059	0.003	0.402	0.020	−0.013	0.143 **	20.612	0.156	0.129
Heart disease	−0.025 **	−6.230	−0.017	−0.033	0.041 **	6.295	0.054	0.028	−0.029 **	−5.482	−0.019	−0.039
Emotional and psychiatric disorders	−0.096 **	−9.710	−0.077	−0.116	−0.370 **	−23.669	−0.340	−0.401	−0.149 **	−11.670	−0.124	−0.174
Chronic pulmonary disease	0.181 **	33.911	0.191	0.170	0.144 **	16.927	0.160	0.127	0.171 **	24.726	0.184	0.157
Stomach or digestive disease	−0.096 **	−14.217	−0.083	−0.109	−0.070 **	−6.448	−0.049	−0.092	−0.108 **	−12.417	−0.091	−0.125
Liver disease	−0.041 **	−6.198	−0.028	−0.053	−0.110 **	−10.565	−0.089	−0.130	0.074 **	8.675	0.091	0.057
Kidney disease	0.023 *	3.363	0.036	0.010	−0.017	−1.592	0.004	−0.039	0.119 **	13.346	0.137	0.102
Asthma	0.053 *	2.016	0.104	0.001	0.132 *	3.196	0.213	0.051	0.105 *	3.124	0.170	0.039
Arthritis or rheumatic disease	0.012	0.928	0.036	−0.013	−0.084 **	−4.252	−0.045	−0.123	−0.056 *	−3.462	−0.024	−0.087
Memory-related diseases	−0.019	−1.455	0.007	−0.045	−0.003	−0.122	0.039	−0.044	−0.030	−1.720	0.004	−0.064
Cancer	0.156 **	16.552	0.174	0.137	0.135 **	9.081	0.164	0.106	0.141 **	11.589	0.165	0.117
Dyslipidemia	−0.057 **	−7.878	−0.043	−0.071	−0.065 **	−5.690	−0.043	−0.088	0.035 **	3.741	0.053	0.017
R^2^	0.371				0.258				0.230			

Ref. = control group; * *p* < 0.05, ** *p* < 0.001.

## Data Availability

Restrictions apply to the availability of these data. Data were obtained from the Health Commission of Gansu Province, and data sharing is not applicable.

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
