# Peer review of "Analysis of Hospitalization Costs in Patients Suffering from Cerebral Infarction along with Varied Comorbidities"

_ijerph, 2022, doi:10.3390/ijerph192215053_

Round 1
Reviewer 1 Report
The paper is interesting, is within the scope of the Journal and in general was well written and organised. Below are some inaccuracies, which need to be corrected or explained:
1. Please change roman indices (I, II, IV etc.) into arabic for the reference citations.
2. lines 47-48 - if you cite costs in dollars, for China also re-calculate it into dollars.
3. line 61 - it should be rather "thorough"
4. line 103 - " Han Chinese" - what does it mean?
5. line 111-118 and many other places in the paper. I understand that Authors provide calculator form Juan to Dollar. But if this is an international journal and the paper is in English, I would suggest to show all costs in Dollars (in the entire manuscript). It will be much easier to compare such a data with data from other papers and similar studies.
6. There is a rule that each table must be self-explanatory. Therefore please clearly indicate currency in each table.
7. Figure 1 is not fully visible. Some part was cut off.
8. In the Conclusion chapter please put some data form the research and not just only general descriptions, which are widely known.
Author Response
Dear reviewer,
Thank you for your letter and the comments concerning our manuscript entitled “Comorbidity is associated with high hospitalization costs with cerebral infarction patients” (ijerph-1982272). Those comments are valuable and very helpful. We have read through comments carefully and have made corrections. Based on the instructions provided in your letter, we uploaded the file of the revised manuscript.
we highly appreciate your time and consideration. In this revision, we use red text to show the modification to the manuscript, here are our responds to the comments you made:
Response to the reviewer's comments:
Question 1. Please change roman indices (I, II, IV etc.) into arabic for the reference citations.
Response: We are grateful for the suggestion. The reference citations have been changed to Arabic numerals. (see lines 286-351).
Question2. lines 47-48 - if you cite costs in dollars, for China also re-calculate it into dollars.
Response: We are grateful for the suggestion. We have re-calculate all the costs in dollars in the manuscript, and the cost of both the table and the text description have been corrected. (see lines 51).
Question3. line 61 - it should be rather "thorough"
Response: Thank you for your careful review. There are two reviewers all put forward suggestions for "through", we comprehensive reviewer suggestions and modified the "through" to " thoroughly". (see lines 74-75).
Question4. line 103 - " Han Chinese" - what does it mean?
Response: We apologize for the language problems in the manuscript. Among the ethnic groups in China, there are usually divided into two main categories, Han nationality and Ethnic minorities. The "Han Chinese" is the most populous ethnic group, we have modified " Han Chinese " to " Han nationality " in the manuscript. (see line 117 and Table 1)
Question 5. line 111-118 and many other places in the paper. I understand that Authors provide calculator form Juan to Dollar. But if this is an international journal and the paper is in English, I would suggest to show all costs in Dollars (in the entire manuscript). It will be much easier to compare such a data with data from other papers and similar studies.
Response: We are grateful for the suggestion and we apologize for this regarding our negligence in this work. According to your advice, we have amended the relevant part in manuscript and show all costs in Dollars in our entire manuscript.
Question 6. There is a rule that each table must be self-explanatory. Therefore please clearly indicate currency in each table.
Response: We are grateful for the suggestion. We clearly indicate currency in each table and figure in the manuscript. (see line 134,Table 2 and line 142,Table 3).
Question7. Figure 1 is not fully visible. Some part was cut off.
Response: We are grateful for the suggestion. We have made further adjustments to Figure 1, so that it can be clearly presented in the paper. (see line152).
Question 8. In the Conclusion chapter please put some data form the research and not just only general descriptions, which are widely known.
Response: We feel great thanks for your professional review work on our article. We tried our best to improve the manuscript. Based on your suggestions, we have made detailed changes to the content of the concluding section of the article and provided relevant research data to support our article and corrected the text description.
We appreciate for your work earnestly, and hope the correction will meet with approval. If our manuscript needs to be revised and improved, we hope to get the opportunity to revise again, and we sincerely hope to get your valuable suggestions to improve the quality of our manuscript. Once again, thank you very much for your comments and suggestions.
Sincerely,
Yongcong Chen

Reviewer 2 Report
ijerph-1982272
Comorbidity is associated with high hospitalization costs with cerebral infarction patients
This paper tries to address something about the hospitalization costs and cerebral infarction patients. However, the novelty is weak but the experiments are good enough to support it for publication. The manuscript must be revised and some of the detailed comments are listed as follows:
1. Title ‘Comorbidity is associated with high hospitalization costs with 2 cerebral infarction patients’ should be changed in a concise way and reflect the main work you done.
2. ‘Conclusions: Comorbidity is significantly associated with high hospitalization costs for cerebral infarction patients. And health-relevant departments should build the prevention and control system to reduce comorbidity risk, improve clinical pathway management of patients, and strength refined cost control management of cerebral infarction from the perspective of comorbidity’ is clear as daylight even if the so-called analysis has not done! Please give reasons to explain the statistical rules!
Author Response
Dear reviewer,
Special thanks to you for your letter and comments concerning our manuscript that entitled “Comorbidity is associated with high hospitalization costs with cerebral infarction patients” (ijerph-1982272). We appreciate the time and effort that the reviewers put into reviewing previous versions of the manuscript. These advices allows us to improve our work.Our point-by-point response to your comments.
Based on your comments, we revised the manuscript and uploaded the revised document. In this revision, we use red text to show the modification to the manuscript, here are our responds to the comments you made:
Response to the reviewer's comments:
Question1.Title ‘Comorbidity is associated with high hospitalization costs with 2 cerebral infarction patients’ should be changed in a concise way and reflect the main work you done.
Response: Our deepest gratitude goes to you for your careful work and thoughtful suggestions that have helped improve this paper substantially. Based on the comments of the reviewers, we have revised the title of the manuscript to " Analysis of hospitalization costs in patients suffering from cerebral infarction along with varied comorbidities ". (see lines2-3).
Question2.Conclusions: Comorbidity is significantly associated with high hospitalization costs for cerebral infarction patients. And health-relevant departments should build the prevention and control system to reduce comorbidity risk, improve clinical pathway management of patients, and strength refined cost control management of cerebral infarction from the perspective of comorbidity’ is clear as daylight even if the so-called analysis has not done! Please give reasons to explain the statistical rules!
Response: In China, government healthcare policies are closely related to the cost of hospitalization for patients. In recent years, China has intensified its healthcare reforms to reduce people's medical costs, implementing an important reform called "single-disease payment" policy: a set of payment standards for each chronic disease, and the government provides medical subsidies according to the type of chronic disease. However, most cerebral infarction patients are combined with a variety of chronic diseases, and comorbidities bring additional medical resources and hospitalization costs, so it is necessary to continuously improve the single-disease payment system, focusing on the problem of "comorbidities" to reduce the hospitalization costs of patients. On the other hand, China government implements the "clinical pathway management" policy, which is an auxiliary measure to the "single-disease payment" policy. According to the cost of drugs, diagnosis, medical services and other expenses of each chronic disease patient during hospitalization, the basic hospitalization cost of the disease is estimated, and then the government formulates corresponding medical security measures. Studies have shown that comorbidities greatly increase the hospitalization cost of cerebral infarction patients. At present, there are few studies on the comorbid mechanism of cerebral infarction, which makes it difficult for the government to better control the hospitalization cost of cerebral infarction based on the perspective of comorbidity. Our study analyzes the types and quantities of comorbidities prone to cerebral infarction and the hospitalization costs to be spent under different combinations of comorbidities, and we hope to provide certain data support for the government to improve the payment policy and management of clinical pathways for cerebral infarction in the future, and strengthen the refined cost control management of cerebral infarction from the perspective of comorbidities, so as to reduce the hospitalization cost of patients.
We tried our best to improve the manuscript and made some changes marked in red in revised paper which will not influence the content and framework of the paper. We hope the correction will meet with approval. If our manuscript needs to be revised and improved, we hope to get the opportunity to revise again, and we sincerely hope to get your valuable suggestions to improve the quality of our manuscript. Once again, thank you very much for your comments and suggestions.
Sincerely,
Yongcong chen

Reviewer 3 Report
Thanks for the paper. Cerebral infarction is a serious condition and its outcome is related closely to quality of life. Health cost due to cerebral infarction can be a significant burden for the patients and their families, particularly in low income countries and communities. Comorbidities can increase hospital length of stay and the costs associated with hospital admission. Understanding of the effects of comorbidities on hospital costs among the patients with cerebral infarction is a good and important topic.
There are a few things the authors need to clarify further:
1 Was "cerebral infarction" the primary diagnosis in the medical records used for this research?
2 Ethical approval: this study was based on medical records but it did use sensitive information such as costs and comorbidities. There should still be an institutional review to be completed to show this research "is out of scope".
3 Data quality of the medical records used for this analysis: what were the proportions of (1) Total hospitalization cost was zero, and (2) Data was missing or wrong, and impossible to correct? Can you demonstrate the data used for this analysis was of good quality?
4 Patients representation: why did you exclude those with a length of stay more than 60 days? What's the implications for the interpretation of your study results?
5 Statistical methods:
5.1 95%CIs should be reported for important outcomes, e.g. Table 2 and Table 3.
5.2 In Table 4, beta and its associated 95%CI should be reported rather than the t value. In addition, the beta itself does not make too much sense to the readers as the hospital related costs were logarithm-transformed. I wonder if there can be some work to transform the beta back so that readers can appreciate the size of the effect of the comorbidities. Also explanation of other variables (particularly the significant ones) are essential in addition to the comorbidity variables.
5.3 This study considered the effects of the number of comorbidities and the comorbidity types/categories separately initially. The multiple linear regression model tried to understand the effects of all these comorbidities, together with length of stay which is another important variable related to the costs. I wonder if Charlson Comorbidity Index can be used in the analysis as this index takes into account the number of comorbidities and the comorbidity categories.
6 Explanation of some variables for the international readers:
6.1 The levels of hospitals in Gansu China: was there a difference in the severity of cerebral infarction? Were there any differences in the charge/cost for the same hospital service?
6.2 The payment mechanism by the government: it would be of good value to provide a general description in the Methods part.
Other minor things are as follows:
Line 20: 'Risk' of comorbidity? 'Risk' is not suitable here as the authors mainly meant the comorbidities that were most common by frequency.
Line 22: 95%CI should also be reported for the cost (also refer to 5.1).
Line 26: 'cerebral infarction' was the study outcome so should not be treated as a 'comorbidity'.
Line 57: add 'and' before 'dyslipidaemia'.
Line 61: Change 'through' to 'thoroughly'.
Line 71: change 'show' to 'showed'.
Line 75: reword this line please.
Line 81: Change '2.2 Enrolled indicators' to '2.2 Variables extracted'.
Lines 84 and 86: Replace 'informations' with 'information'.
Lines 91-92: Reword the two lines. Change 'normal distribution' to 'normally distributed'.
Lines 104-105: Change '...none of the comorbidity' to 'none of the comorbidities'.
Table 1: This table needs to be reformatted to make the structure clear, e.g. 'Han Chinese' should indent under 'Ethnicity'. 'Total Y'?
Line 110: '3.2. The status of hospitalization costs in different numbers of comorbidity' can be replaced with '3.2. Association between hospitalization costs and the number of comorbidities'.
Table 2: I wonder if there should be a trend test.
Line 123: This line needs to be reworded. Perhaps something like: '... the number of patients comorbid with hypertension...'
Line 129 - Table 3: I wonder what happened if patients had multiple comorbidities? Were these categories/rows exclusive to each other? Or in other words, could one patient appear in more than one category of comorbidities?
Figure 1: Not shown properly and this needs to be corrected.
Table 4: No Chinese words please.
Line 154: Change 'researches' to research'.
Lines 152-232: The 'Discussion' part needs to be reworded - the grammar has to be improved.
Line 236: Replace 'resuce' with 'reduce'.
Author Response
Dear reviewer,
Special thanks to you for your letter and comments concerning our manuscript that entitled “Comorbidity is associated with high hospitalization costs with cerebral infarction patients” (ijerph-1982272). We appreciate your professional guidance of our manuscript. These comments were very helpful to us, and they pointed out the deficiencies of our manuscripts, and where we did not do enough. It plays an important role in guiding our future in-depth research on cerebral infarction and comorbidities and helps us further improve our research.
We have made corresponding corrections according to your guidance and suggestions, and we have left traces of changes in the revised manuscript. In this revision, all changes to our manuscript are shown in red text in the document, and here are our responds to the comments you made:
Response to the reviewer's comments:
Question1. Was "cerebral infarction" the primary diagnosis in the medical records used for this research?
Response: Thank you for your careful review. The "cerebral infarction" is the primary diagnosis in the medical records in this study.
Question2. Ethical approval: this study was based on medical records but it did use sensitive information such as costs and comorbidities. There should still be an institutional review to be completed to show this research "is out of scope".
Response: We are grateful for the suggestion. This research has been reviewed by the ethics committee of Lanzhou University, with an approval number and institutional review documents. But at the moment we were in a difficult situation, because of the sudden outbreak of the COVID-19 in Lanzhou City, Gansu Province, Lanzhou university was closed for management, we were isolated in a separate room, the review documents were sealed in the office building, and we could not obtain the institutional review certificate at present. If this is possible, we would like to have the opportunity to submit the ethics review report by one month, and we greatly appreciate your understanding.
Question3. Data quality of the medical records used for this analysis: what were the proportions of (1) Total hospitalization cost was zero, and (2) Data was missing or wrong, and impossible to correct? Can you demonstrate the data used for this analysis was of good quality?
Response: We are grateful for the suggestion. The data of this study were extracted by professionals of the Health Commission of Gansu Province to ensure the standardization of data extraction. The inclusion criteria for study data were rigorous. Among the 84726 patients, 8163 were excluded, and 76563 patients with cerebral infarction were finally included
Exclusion Criteria as follows:
(1) Total hospitalization cost was zero——It is not logical for the hospitalization cost to be zero, so the relevant data is excluded to ensure the standardization of the data, and a total of 259 patients are excluded, accounting for 0.31%.
(2) Data was missing or wrong, and impossible to correct——Excluding samples with missing information or errors to ensure the integrity of the data, a total of 4113 patients were excluded, accounting for 4.85%.
(3) Length of hospital stay was less than 1 day or more than 60 days——A length of hospital stay of 1 day or a length of hospital stay of more than 60 days skewed the study data and did not meet study criteria, thus excluding both types of patients. A total of 3791 patients were excluded, accounting for 4.47%.
In summary, this study ensures the standardization and integrity of the sample and the good quality of the data.
Question4. Patients representation: why did you exclude those with a length of stay more than 60 days? What's the implications for the interpretation of your study results?
Response: We are grateful for the comment. China's medical policy is related to the number of days of hospitalization and medical insurance reimbursement policy, and the government promotes the flow of medical resources by limiting the number of days of hospitalization. The number of hospitalization days is usually about 2 weeks, and some medical expenses cannot be reimbursed by medical insurance after 30 days, so the number of hospitalization days exceeding 60 days is abnormal, which may be due to errors in records or other special circumstances. In order to ensure the standardization and quality of the study, we excluded patients who were hospitalized for more than 60 days.
Question5. Statistical methods:
5.1 95%CIs should be reported for important outcomes, e.g. Table 2 and Table 3.
Response: We are grateful for the suggestion. In Table 2, We analyzed hospitalization costs of cerebral infarction patients based on the different number of comorbidities, The cost of hospitalization increases with the number of comorbidities; Kruskal-Wallis H test results were statistically significant, H=404.506, P<0.001, which can support our research viewpoint. So we used the Kruskal-Wallis test and therefore did not report 95% CI. In Table 3, we mainly analyze the status of hospitalization costs in different type of comorbidity. We used a descriptive analysis approach, so 95% CI results were not reported in the table.
5.2 In Table 4, beta and its associated 95%CI should be reported rather than the t value. In addition, the beta itself does not make too much sense to the readers as the hospital related costs were logarithm-transformed. I wonder if there can be some work to transform the beta back so that readers can appreciate the size of the effect of the comorbidities. Also explanation of other variables (particularly the significant ones) are essential in addition to the comorbidity variables.
Response: We are grateful for the professional suggestion. We have improved the table 4 and revised and supplemented the table. We supplemented beta and its associated 95%CI and reported both the t value.The significant variables (p-values) are represented by symbols, with * representing P<0.05 and ** representing P<0.001. (see lines 164).
5.3 This study considered the effects of the number of comorbidities and the comorbidity types/categories separately initially. The multiple linear regression model tried to understand the effects of all these comorbidities, together with length of stay which is another important variable related to the costs. I wonder if Charlson Comorbidity Index can be used in the analysis as this index takes into account the number of comorbidities and the comorbidity categories.
Response: We are grateful for the suggestion. Charlson Comorbidity Index (CCI is an great comorbidity scoring system, and your suggestions have brought important guidance and ideas to our future research on cerebral infarction comorbidities. After trying the CCI method based on the existing data, we found that the current data we obtained have limitations. It is difficult to establish weights based on the disease codes of comorbidities to obtain the sum of the weights of all comorbidities, which makes us unable to calculate the total CCI score of cerebral infarction patient. We feel sorry that we did not provide enough information about it, but our team still regards the comorbidity and cost of cerebral infarction as the focus of future research, and we hope to use the Charlson Comorbidity Index for in-depth research in future research.
Question6. Explanation of some variables for the international readers:
6.1 The levels of hospitals in Gansu China: was there a difference in the severity of cerebral infarction? Were there any differences in the charge/cost for the same hospital service?
Response: We are grateful for the suggestion. The levels of hospitals in Gansu China:
The situation of each patient with cerebral infarction inevitably has certain differences, and the severity of cerebral infarction is not clearly graded during hospital treatment, and the severity of cerebral infarction is not entered into medical records, so we unable to know the exact severity. There is less difference in the charge/cost for the same hospital service. The price of each drug, the price of each diagnosis and the cost of medical services are clear, while records of service charges are kept and hospital charges are checked annually by government agency staff. The doctor will give different treatment plans according to the patient's situation, and the difference in treatment plan makes the cost of hospitalization different. In addition, different patients' medical insurance also makes the final hospitalization expenses different.
6.2 The payment mechanism by the government: it would be of good value to provide a general description in the Methods part.
Response: We are grateful for the suggestion. In the methods section, we mainly wrote about data sources, indicator inclusion, and research methods; Considering the importance of a general description of government payment mechanisms, we have added an introduction to the "introduction" to the introduction of government payment mechanisms. In the Chinese government's payment mechanism, the medical service system implements hierarchical diagnosis and treatment, and at the same time carries out single-disease cost accounting based on clinical pathway management, and clarifies the cost pricing of various equipment inspections, drugs and diagnosis and treatment services. The research on comorbidities is closely related to the government payment mechanism and the hospitalization cost of patients, and the government payment mechanism is improved from the perspective of comorbidities to reduce the hospitalization cost of patients with cerebral infarction. (see lines 53-62).
Question7. Other minor things are as follows:
1.Line 20: 'Risk' of comorbidity? 'Risk' is not suitable here as the authors mainly meant the comorbidities that were most common by frequency.
Response: We are grateful for the suggestion. Your suggestions help us refine the details of our manuscript. The use of the word "risk" in the most common comorbidities of cerebral infarction is really inappropriate. What we want to say is that among the chronic diseases studied, hypertension is the chronic disease with the highest frequency of comorbidities of cerebral infarction, so we changed the sentence to " Regarding the incidence of varied chronic disease comorbidities concomitant with cerebral in-farction, hypertension was reported as the highest, followed by heart disease and chronic pul-monary disease" in the manuscript. (see lines22-24).
2.Line 22: 95%CI should also be reported for the cost (also refer to 5.1).
Response: Thank you for your careful review. We have reported the relevant 95% CI in the article table. (see line155).
3.Line 26: 'cerebral infarction' was the study outcome so should not be treated as a 'comorbidity'.
Response: We are grateful for the suggestion. Our expression was not clear enough, and we have revised the cerebral infarction comorbidity part. Cerebral infarction was only a study result, not as a 'comorbidity'. (see lines27-30).
4.Line 57: add 'and' before 'dyslipidaemia'.
Response: We are grateful for the suggestion. we add 'and' before 'dyslipidaemia' in the manuscript. (see line60).
5.Line 61: Change 'through' to 'thoroughly'.
Response: We are grateful for the suggestion. We have modified this expression and we change 'through' to 'thoroughly' in the paper. (see line65).
6.Line 71: change 'show' to 'showed'.
Response: We are grateful for the suggestion. We have modified 'show' to 'showed' in the sentence. (see line75).
7.Line 75: reword this line please.
Response: We have reword this line in the manuscript:
(1) According to the International Classification of Diseases, patients with the primary diagnostic disease code I63.900 were included. (see lines 80-81).
8.Line 81: Change '2.2 Enrolled indicators' to '2.2 Variables extracted'.
Response: We are grateful for the suggestion. We have modified '2.2 Enrolled indicators' to '2.2 Variables extracted'. (see line 87).
9.Lines 84 and 86: Replace 'informations' with 'information'.
Response:We are grateful for the suggestion. We have replaced 'informations' with 'information'. (see line 90 and 92).
10.Lines 91-92: Reword the two lines. Change 'normal distribution' to 'normally distributed'.
Response: We are grateful for the suggestion. We have changed this sentence to: The hospitalization costs, drug costs, and diagnostic costs are logarithmic and distributed in an approximately normal manner. (see lines 97- 98).
11.Lines 104-105: Change '...none of the comorbidity' to 'none of the comorbidities'.
Response: We are grateful for the suggestion. We have Changed '...none of the comorbidity' to 'none of the comorbidities'. The changed sentence is as follows: 12,519 patients (16.35%) did not have any comorbidities, while 64,044 patients (83.64%) suffered with comorbidities. The risk of comorbidity for patients aged 60-80 years was the highest, followed by 40-60 years old (2=0.080, P<0.001), males had a significantly higher risk of stroke than females (2=0.023, P<0.001). (see lines 109- 113).
12.Table 1: This table needs to be reformatted to make the structure clear, e.g. 'Han Chinese' should indent under 'Ethnicity'. 'Total Y'?
Response: We are grateful for the suggestion. We have reformatted the table to make the table clearer; "Han Chinese" has been indented under "race". "Total Y" is a typo in the article and has been revised.
13.Line 110: '3.2. The status of hospitalization costs in different numbers of comorbidity' can be replaced with '3.2. Association between hospitalization costs and the number of comorbidities'.
Response: We are grateful for the suggestion. We have replaced '3.2. The status of hospitalization costs in different numbers of comorbidity' to '3.2. Association between hospitalization costs and the number of comorbidities'. (see line 115)
14.Table 2: I wonder if there should be a trend test.
Response: We are grateful for the suggestion. We sincerely thank the reviewer for careful reading.In Table 2, we use the Kruskal-Wallis H test for methodology. We all agree that the trend test will make the relationship between the number of comorbidities and the cost of hospitalization clearer. In this study, we tried to explore the trend test, we are sorry to not get the exact trend test results, due to some limitations of the data, our trend test was not successful. We checked the Kruskal-Wallis H test results again, and the results can support our research thesis that there are differences in hospitalization costs under different comorbidities, and the number of comorbidities affects the hospitalization costs of patients with cerebral infarction. Our research still continues, and we will continue to explore, and we hope to further study the trend relationship between the number of comorbidities and hospitalization cost.
15.Line 123: This line needs to be reworded. Perhaps something like: '... the number of patients comorbid with hypertension...'
Response: We are grateful for the suggestion. We have reworded this sentence to ' Among the types of comorbidity, the number of patients comorbid with hyper-tension (n=45,626) was the highest '. (see lines 127-128).
16.Line 129 - Table 3: I wonder what happened if patients had multiple comorbidities? Were these categories/rows exclusive to each other? Or in other words, could one patient appear in more than one category of comorbidities?
Response: The number of comorbidities that affect the cost of hospitalization is increased, and the more medical resources are required for patients. For example, when cerebral infarction is complicated by hypertension, drugs and treatments that increase hypertension are required; At the same time, hypertension and heart disease require additional examinations and medications, increased consumption of medical resources, and increased hospitalization costs. The chronic disease categories associated with cerebral infarction affect each other. For example, cerebral infarction combined with hypertension, diabetes, dyslipidemia is easy to accompany the occurrence, from the analysis of deep-seated causes, combined chronic diseases will have some of the same risk factors exposed, chronic diseases affect each other. One patient can appear in more than one category of comorbidities.
17.Figure 1: Not shown properly and this needs to be corrected.
Response: We are grateful for the suggestion. We have corrected the Figure 1 in the paper. (see line144, Figure 1).
18.Table 4: No Chinese words please.
Response: We feel sorry for our carelessness. In our resubmitted manuscript, t we have revised the Chinese words. Thanks for your correction.
19.Line 154: Change 'researches' to research'.
Response: We are grateful for the suggestion. We have changed 'researches' to research'.
(see line159).
20.Lines 152-232: The 'Discussion' part needs to be reworded - the grammar has to be improved.
Response: We are grateful for the suggestion. We have corrected the content and syntax of the discussion section.
21.Line 236: Replace 'resuce' with 'reduce'.
Response: We are grateful for the suggestion. We have replaced 'resuce' with 'reduce'.
(see line253).
We really appreciate your professional advice. If there are any other imperfections in our article, we hope to get your guidance again, and we will continue to revise and improve the manuscript. Thanks again for your time and consideration.
Sincerely,
Yongcong chen
